# Lmx1a-Dependent Activation of miR-204/211 Controls the Timing of Nurr1-Mediated Dopaminergic Differentiation

**DOI:** 10.3390/ijms23136961

**Published:** 2022-06-23

**Authors:** Salvatore Pulcrano, Roberto De Gregorio, Claudia De Sanctis, Laura Lahti, Carla Perrone-Capano, Donatella Ponti, Umberto di Porzio, Thomas Perlmann, Massimiliano Caiazzo, Floriana Volpicelli, Gian Carlo Bellenchi

**Affiliations:** 1Institute of Genetics and Biophysics “A. Buzzati-Traverso”, National Research Council (C.N.R.), 80131 Naples, Italy; salvatore.pulcrano@igb.cnr.it (S.P.); roberto.degregorio@einsteinmed.edu (R.D.G.); claudia.desanctis86@gmail.com (C.D.S.); diporzio@igb.cnr.it (U.d.P.); massimiliano.caiazzo@unina.it (M.C.); 2Department of Pharmacy, School of Medicine and Surgery, University of Naples Federico II, 80131 Naples, Italy; perrone@unina.it; 3The Ludwig Institute, Department of Cell and Molecular Biology, Karolinska Institute, 17177 Stockholm, Sweden; laura.lahti@ki.se (L.L.); thomas.perlmann@ki.se (T.P.); 4Department of Medical-Surgical Sciences and Biotechnologies, University of Rome Sapienza, 040100 Latina, Italy; donatella.ponti@uniroma1.it; 5Department of Molecular Medicine and Medical Biotechnology, University of Naples Federico II, 80131 Naples, Italy; 6Department of Pharmaceutics, Utrecht Institute for Pharmaceutical Sciences (UIPS), Utrecht University, Universiteitsweg 99, 3584 CG Utrecht, The Netherlands; 7IRCCS Fondazione Santa Lucia, 00179 Rome, Italy

**Keywords:** microRNA, dopamine, Nurr1, Lmx1a

## Abstract

The development of midbrain dopaminergic (DA) neurons requires a fine temporal and spatial regulation of a very specific gene expression program. Here, we report that during mouse brain development, the microRNA (miR-) 204/211 is present at a high level in a subset of DA precursors expressing the transcription factor Lmx1a, an early determinant for DA-commitment, but not in more mature neurons expressing Th or Pitx3. By combining different in vitro model systems of DA differentiation, we show that the levels of Lmx1a influence the expression of miR-204/211. Using published transcriptomic data, we found a significant enrichment of miR-204/211 target genes in midbrain dopaminergic neurons where Lmx1a was selectively deleted at embryonic stages. We further demonstrated that miR-204/211 controls the timing of the DA differentiation by directly downregulating the expression of Nurr1, a late DA differentiation master gene. Thus, our data indicate the Lmx1a-miR-204/211-Nurr1 axis as a key component in the cascade of events that ultimately lead to mature midbrain dopaminergic neurons differentiation and point to miR-204/211 as the molecular switch regulating the timing of Nurr1 expression.

## 1. Introduction

Although dopaminergic neurons (DAn) are heterogeneously distributed in the mammalian brain, the main dopamine (DA) source resides in the ventral midbrain. Those midbrain dopaminergic neurons (mDAn) participate—among their many functions—in the control of voluntary movements, reward, and decision-making [1,2].

mDAn are organized into two distinct nuclei with peculiar functional features: the substantia nigra (SN) and the ventral tegmental area (VTA) [3]. Patients with Parkinson’s disease (PD) and PD animal models manifest specific degeneration of mDAn in the SN while alterations of those neurons in the VTA are linked to Attention Deficit Hyperactivity Disorder (ADHD), stress-related psychopathologies, schizophrenia, and addiction [1,4,5].

During development, mDAn arise from the floor plate (FP) through the sequential activation of distinctive signals and peculiar transcription factors (TFs). Among these, Lmx1a plays a key role in committing neuroblasts toward the mDAn fate delimiting the DA domain [6,7]. Loss of Lmx1a causes a reduction in the number of mDAn in the mouse ventral midbrain mainly due to a proliferative deficit in the DA precursor population, but it does not result in a complete loss of mDAn, likely because of a partial compensation operated by its homolog Lmx1b [7,8,9]. Indeed, Lmx1a and Lmx1b have partially overlapping roles. Lmx1b is essential in the regulation of dorso-ventral patterning of the limbs and organ development, including the anterior segment of the eye, and in the midbrain, influences the development of DA and non-DA compartments such as the oculomotor nucleus (OMN) and the red nucleus (RN). Lmx1a starts being expressed in the progenitor zone of the mouse ventral midbrain from embryonic (E) day 9.5 in a spatiotemporal pattern correlated with the onset of midbrain DA neurogenesis [6,10,11]. Several studies, mainly in chick and embryonic stem cells (ESC), have suggested that Lmx1a mediates neurogenesis and the transition from the proliferative zone to the intermediate zone, through the activation of another homeobox gene, Msx1, and, in turn, Ngn2 [6,12,13]. At the same time, Lmx1a, by negatively regulating Lim1 and Nkx6-1 via Msx1, defines DA borders by inhibiting alternative neuronal fate [6,14]. In post-mitotic mDAn, Lmx1a/b directly regulates the expression of the dopamine transporter (DAT) and the vesicular monoamine transporter 2 (Vmat2), genes necessary for mDAn terminal differentiation and markers of mature mDAn [15,16,17]. According to recent findings, Lmx1a/b expression is also maintained throughout adulthood, and, by regulating autophagy and mitochondrial metabolism, provides protection against cellular stress [18,19]. For all these reasons, Lmx1a is effectively used in combination with other TFs to improve DA differentiation in vitro and in vivo from different cellular systems [15,20,21,22,23].

The orphan nuclear receptors Nurr1, also named Nr4a2, and Nur77, named Nr4a1, are both expressed in DA neurons. While Nur77 is closely associated with dopamine neurotransmission in the mature brain [24], Nurr1 is mandatory during development for the terminal differentiation of mDAn. Its expression starts at E10.5, one day after Lmx1a activation, in a more ventral portion of the Lmx1a positive domain [6,9]. Nurr1 is required for post-mitotic differentiation, maturation, and survival of mDAn through its action in regulating cell cycle progression and the expression of mature mDAn specific genes, such as tyrosine hydroxylase (*Th*), *Pitx3*, *Vmat2*, and *DAT* or genes involved in cell survival, i.e., *Ret* and the brain-derived neurotrophic factor (*Bdnf*) [25,26,27,28,29,30,31,32]. Remarkably, a loss of Nurr1 results in the depletion of the mDAn neuronal population [33]. Thus, fine-tuning Nurr1 or other key DA TFs has been considered a potential approach to promote DA development.

MicroRNAs (miRNAs) are post-transcriptional regulators of gene expression that act as crucial players in brain development, including the midbrain [34,35], and have recently emerged as being able to coordinate early differentiation processes and promote in vitro DA differentiation [36,37]. Among these, miR-34b/c and miR-135a2 regulate proliferative signals via the Wnt1 and Lmx1b pathways modulation in the midbrain [36,38]. Interestingly, miR-204/211 is upregulated in the SN of PD patients, in the serum of patients with sporadic PD, and in animal and cellular models of PD [39,40,41]. These data highlight its possible involvement in the etiopathogenesis of dopaminergic-related disorders and points to the modulation of miR-204/211 to increase mDAn survival.

mmu-miR-204-5p and mmu-miR-211-5p (hereafter renamed miR-204 and miR-211, respectively) are two homologous microRNAs, with the first one located intronically to the Trpm3 gene while miR-211 is hosted in the Trpm1 gene. They share the same identical seed sequence and have only a single nucleotide difference in their mature forms. Because of this, they are functionally and structurally identical and are hereafter referred to as miR-204/211. miR-204/211 is widely expressed in neuronal tissues, including the cerebral cortex, hippocampus, eye, and choroid plexus [35,42,43,44,45,46,47].

MiR-204, together with miR-93 and miR-302d, has been shown to regulate the expression of Nurr1 in rat DA neurons [48]. In the same way, miR-204 seems to also regulate the expression of Nur77 in human brain microvascular endothelial cells [49].

Herein we investigate the link between miR-204/211 and the DA TFs Lmx1a and Nurr1. We demonstrate that Lmx1a induces the activation of miR-204/211 and, in turn, the latter regulates Nurr1 expression, influencing mDAn development. Thus, we believe that the Lmx1a/miR-204/211 axis controls the timing of differentiation of midbrain precursor cells, and its modulation could also be used to improve in vitro generation of mDAn, useful for the development of a model system or cell therapy.

## 2. Results

### 2.1. miR-204/211 Is Expressed in Mesencephalic DA Progenitor Cells

To evaluate miR-204/211 expression in mDAn, we compared its expression in the cortex and midbrain of E14.5 and adult mice using qPCR. miR-204/211 shows enrichment in the midbrain compared to the cortex both in E14.5 and adult mice (Figure 1a,d). A similar profile is observed by measuring the expression levels of its host gene, Trpm3 (Figure 1b,e), while Trpm1, the host gene for miR-211, is enriched only in the E14.5 midbrain (Figure 1c,f).

To characterize whether miR-204/211 is ubiquitously expressed in all DA cells during development, we analyzed its expression in isolated mDAn, expressing Lmx1a, a marker of DA progenitors or early neurons, and Th or Pitx3, markers of differentiated mDAn. For this purpose, we purified GFP^+^ neurons from the Lmx1a-GFP and Pitx3-GFP knock-in mouse lines, where the mDA neurons Lmx1a^+^ or Pitx3^+^ were exclusively labelled by the GFP reporter, and from the Th-GFP transgenic mouse line, where the GFP was located downstream of the Th promoter [8,50,51].

We observed that miR-204/211 is enriched in GFP^+^ cells derived from Lmx1a-GFP mice at E12.5 to E14.5 (Figure 1g) but is significantly reduced in GFP^+^ (vs. GFP^−^) cells purified from Th-GFP and Pitx3-GFP mice at E12.5 to E15.5 (Figure 1h,i). During development, in the ventral midbrain, Th and Pitx3 domains appear as subdomains of Lmx1a^+^ cells from which they differentiate. Thus, the enrichment of miR-204/211 in Lmx1a^+^ DA progenitors—but not in differentiating (Pitx3^+^) or differentiated (Th^+^) mDAn, suggesting miR-204/211 downregulation in postmitotic mDAn precursors—could identify the specific Lmx1a^+^ sub-population involved in temporal and/or spatial regulation of the DA domain.

### 2.2. The Levels of Lmx1a Influence the Expression of miR-204/211 and Its Predicted Target Genes

Since only Lmx1a^+^ cells are enriched for miR-204/211, we hypothesized that miR-204/211 plays a role in the Lmx1a-regulated commitment of mDAn and, therefore, that Lmx1a could influence its expression. To confirm this hypothesis, we used GFP^+^ and GFP^−^ cells isolated from E12.5 Lmx1a^+/GFP^ and Lmx1a^GFP/GFP^ mouse embryos corresponding to Lmx1a heterozygous (Lmx1a^+/−^) and knock-out (Lmx1a^−/−^) animals, respectively. Interestingly, in such a context, miR-204/211 was significantly enriched in GFP^+^ cells derived from Lmx1a^+/GFP^ embryos compared to those derived from Lmx1a^GFP/GFP^ indicating that, in the absence of Lmx1a, the expression of the miR-204/211 is also reduced (Figure 2a,b). As a control, we analyzed the expression of two other miRNAs, miR-218 and miR-9. The first is known to be enriched in mDAn [52] while the second is broadly expressed in the brain [53,54]. Differently from miR-204/211, the expression of both miR-218 and miR-9 was not affected by the deletion of Lmx1a (Figure 2a,b), thus suggesting that miR-204/211 expression may be directly affected by Lmx1a.

To validate this hypothesis, we transfected the mes-cmyc-A1 cell line, an in vitro model of mDAn [55], with an inducible Lmx1a-Ires-GFP vector or with an empty Ires-GFP vector as a control. GFP^+^ cells, overexpressing Lmx1a or not, were FACS-purified and analyzed for the expression of miR-204/211. As expected, we found an enrichment of miR-204/211 in the Lmx1a-Ires-GFP^+^ cells but not in the control GFP^+^ cells (Figure 2c), supporting our hypothesis that miR-204/211 expression is affected by Lmx1a activity.

To corroborate this idea, we analyzed previously published data obtained from E15.5 DA neurons derived from Lmx1a/b double KO (Lmx1a/b-KO) [56]. Thus, we scanned the entire list of differentially expressed genes (DEG) between KO vs. WT for the presence of putative binding sites for miR-204/211 by using the DIANA-microT-CDS prediction tool [57]. Out of 224 DEG, 27 (12%) were predicted to be targets for the miR-204/211 (Figure 2d). Interestingly, 66.7% of these (18 genes) are upregulated in the Lmx1a/b-KO compared to the WT cells, while when looking at the unpredicted not targeted genes, they are equally distributed between down and upregulated (48.2% positive fold change; 51.8% negative fold change) (Figure 2e).

### 2.3. miR-204/211 Expression Is Regulated by Lmx1a

To confirm the role of Lmx1a in the regulation of miR-204/211 expression during mDAn development, we overexpressed the TF Lmx1a and Nurr1 alone or in combination, in four different in vitro models of mDAn differentiation. These are mes-c-myc-A1, E12.5 midbrain primary cultures (mE12.5-PCs), epiblast-derived stem cells (epiSCs), and induced DA neurons (iDAn) obtained from MEF cells reprogrammed by the co-transfection of the TF Ascl1 with Nurr1 and/or Lmx1a. The overexpression of Lmx1a alone, or in combination with Nurr1, promoted the up-regulation of miR-204/211 (Figure 3a–d) in all in vitro model systems, while the overexpression of Nurr1 alone, which is essential to induce Th expression and generate functional iDAn cells in vitro (Figure 3e–h), does not affect miR-204/211 expression (Figure 3a–d). In a similar way, the expression of Trpm3, the miR-204/211 host gene, was upregulated in epiSC in the presence of Lmx1a alone or in combination with Nurr1, instead of in iDAn in the presence of Ascl1 and Nurr1 (Figure 3i–l), reflecting a possible partial co-regulation of the miRNA with its host gene.

These results confirm that miR-204/211 is regulated, directly or indirectly, by Lmx1a and indicate that miR-204/211 is not required for Th expression and is not influenced by Nurr1- mediated DA differentiation.

Similar results were observed by using the human DA cellular system SHSY-5Y and the non-neuronal HeLa cells. The expression of Lmx1a in both cell lines promotes the upregulation of miR-204/211 suggesting that the same Lmx1a/miR-204/211 regulation occurs in humans as well (Appendix A).

### 2.4. The Lmx1a-miR-204/211 Axis Controls the Timing of Midbrain Precursors Differentiation

Lmx1a controls the differentiation of DA progenitors from the FP, promoting the transition from the progenitors’ domain to more ventral and differentiating domains. Its expression maintains progenitor proprieties and neurogenic potential, regulating proneural genes. Similarly, miR-204/211 has been reported to maintain adult neural stem cells (NSCs) in an undifferentiated state but primed for neurogenesis [45] in order to control the balance between self-renewal and differentiation and to modulate the gene expression programs that mediate eye development [42,43]. Thus, we hypothesize that Lmx1a may control, at least in part, progenitor identity via miR-204/211.

To further explore this idea, we evaluated the enrichment for miR-204/211 targets amongst 443 genes specifically expressed in FP at E10.5 of mouse embryos [58], where miR-204/211 was predicted among the top 10 in terms of mRNA–miRNA pairing. By using the TargetScan algorithm, we identified 52 genes (11.2%) as highly predicted targets of miR-204/211 (FDR = 7.2 × 10^−8^ Appendix A), while a reduced number of miR-204/211 targets was identified in ventral lateral genes (Appendix A). Following the same approach, we also observed that miR-204/211 is enriched in NSCs and might target 29,7% of NSC genes (FDR = 7.1 × 10^−7^; Appendix A) [45]. Altogether, these results point to the potential role of miR-204/211 as a regulator of the DA differentiation process. To select key miR-204/211 targets in the early phases of DA differentiation, we combined these data with our previously published array data obtained from epiSCs differentiated towards the DA phenotype [36]. Here, by using DIANA-microT-CDS and TargetScan algorithms, we selected a set of FP and NSC genes being predicted as a miR-204/211 targets and showing an opposite expression profile to that of miR-204/211 during the in vitro epiSC to mDAn differentiation (Figure 4a; see Section 4 for details). Following this approach, we identified 14 FP genes (Nurr1, Gdnf, Tcf12, Rasgef1b, Itpr1, Samd5, Wnt4, Arx, Nr3c1, Glis3, Tmem64, Adamts9, Gcnt2, and Fam43a) and 8 NSC genes (Nurr1, Elavl2, Elavl4, Plxna2, Sox4, Sox11, Khdrbs1, and Sox4) (Figure 4b–d and Appendix A). Interestingly, Nurr1 was identified as the only common miR-204/211 target between the FP and NSC genes.

Through further analysis, we observed the existence of a temporal correlation during epiSC DA differentiation between the expression of Lmx1a, miR-204/211, Nurr1, and Pitx3. Indeed, the initial phases of differentiation, between in vitro day 5 and day 9, were characterized by an increased expression of Lmx1a and miR-204/211 (Figure 4e and Appendix A). In this window, the expression of Nurr1 was maintained at a low level, as well as that of its known target Pitx3. Conversely, regarding the differentiation process, between days 9 and 14, we observed a further increase in Lmx1a expression, a decrease in miR-204/211, and a progressive upregulation of Nurr1 and Pitx3 mRNAs (Figure 4e). These data suggest that Lmx1a initially regulates the expression of miR-204/211 that, in turn, influences the timing of Nurr1 and Pitx3/Th expression and mDAn differentiation.

Interestingly, Nurr1 and Gdnf are the only genes with a clear opposite expression to that of miR-204/211, both at day 9 and day 14 of the epiSC DA differentiation. They are low during the initial phases of differentiation but increased later during differentiation (Figure 4d and Appendix A). These trends were similar but less evident for a few other identified genes (Elavl2, Sox4, Plxna2, and Rasgef1b) (Figure 4d and Appendix A).

These observations highlight the importance of finely regulated expression timing for miR-204/211, and in turn, its target genes, during early mDAn development, and suggest a role of this miRNA in defining specific mDAn subpopulations (Figure 4f). In this context, considering the screening results, Nurr1 came out as a strong potential candidate for further investigation.

### 2.5. miR-204/211 Regulates Nurr1 Expression and Influences DA Differentiation

To investigate whether the Lmx1a-miR-204/211 axis could have a role in controlling Nurr1 expression and timely regulating the mDAn differentiating program, we used the computational prediction tools TargetScan 7.2 (http://www.targetscan.org/vert_72/ accessed on 20 June 2022) and DIANA-microT-CDS (http://diana.imis.athena-innovation.gr/DianaTools/ accessed on 20 June 2022). We found that miR-204/211 has a conserved 7mer-m8 binding site on the 3′UTR of both mouse Nurr1 transcripts (Figure 5a). To validate this prediction and investigate whether miR-204/211 could really act as a post-transcriptional regulator of Nurr1, we performed a luciferase reporter assay in HeLa cells by cloning 1233 bp of Nurr1 3′UTR (corresponding to mouse Ch2: 56,997,526 to 56,998,759), containing the predicted miR-204/211 binding site, downstream of the luciferase reporter gene stop codon in the pMIR-Report. The Nurr1-3′UTR (WT) or the same sequence deleted in the binding site for miR-204/211 (MUT) were co-transfected in combination with a miR-204/211 expressing vector. As shown in Figure 5b, 48 h after transfection, miR-204/211 was able to significantly reduce luciferase activity by approximately 33%. This effect was absent when the miR-204/211 binding site on the Nurr1-3′UTR was deleted and specific to the miR-204/211, since miR-218, which is not predicted as a potential regulator of Nurr1 expression, did not affect the luciferase activity.

These data show that miR-204/211 is able to bind the 3′UTR of Nurr1 and downregulate its expression and are in accordance with previous reports [48,49].

Interestingly, the miR-204/211-mediated downregulation of Nurr1 affects the expression of specific Nurr-1 downstream targets such as Th, Vamt2, and DAT that were significantly reduced upon miR-204/211 overexpression in differentiated mes-c-myc-A1 cells (Figure 5c–f). This observation suggests that miR-204/211 overexpression keeps the cells in a more undifferentiated stage.

## 3. Discussion

During midbrain development, changes in gene expression result in the transition from the progenitor to the differentiative phase that anticipates the generation of distinct neuronal populations. The process is finely regulated in space and time since inappropriate timing could result in premature differentiation with consequently reduced numbers of mature neurons.

A specific set of transcription factors acts in a concerted temporal fashion to promote cell cycle exit and initiate DA differentiation. In this context, the TF Lmx1a regulates, among other functions, the neurogenic potential of DA precursors, controlling the expression of floor plate progenitor genes. Thus, by regulating the proneural factor Ngn2 and inhibiting alternative factors specific to other neuronal types, Lmx1a determines the DA progenitor identity. Indeed, at E12.5, the apical portion of the Lmx1a domain maintains, up to birth, progenitor proprieties as demonstrated by the positive staining for Ngn2, Ascl1, Ki67, Sox2, and Nestin, while in the same domain, the expression of the differentiating factor Nurr1, which marks mainly post-mitotic mDAn [6,9,59], is kept repressed. Together with Pitx3 and before the Nurr1 expression, Lmx1a appears when the transition from proliferation to differentiation occurs, playing an essential role in this process.

Alongside transcription factors, microRNAs emerged as a fine-tuner of specific gene networks responsible for the differentiation process for most neuronal types. In oligodendrocytes, the homeobox gene Sox10 is essential for the differentiation and expression of miRNAs with relevance in oligodendrocytes development, including miR-338, miR-335, and miR-155 [60,61]. In DA neurons, miR-34b/c and miR-135 instead regulate the expression of key components of the Wnt pathway facilitating the expansion of the progenitor pool, just before the start of progenitor differentiation [36,38]. Similarly, in motor neurons, miR-218 is essential to repressing alternative differentiation programs during motor neuron differentiation, facilitating the proper establishment of the neuro-muscular junction [62].

In a similar way, miR-204/211 mediates retina development and lens formation by controlling the Meis2/Pax6, as well as sustaining the undifferentiated state of the neuronal stem cells [42,45].

Herein, we described a new regulatory axis that starts with Lmx1a and controls the timing for mDAn differentiation through the negative regulation of the post-mitotic transcription factor Nurr1, via miR-204/211. Thus, in the midbrain, Lmx1a mediated the expression of miR-204/211, preventing the start of terminal differentiation, driven by Nurr1, which in turn promotes the expression of terminal DA markers. In addition, the enriched expression of miR-204/211 in Lmx1a^+^ cells, but not in Pitx3^+^ or Th^+^ cells, reveals a subpopulation of primed DA neurons similar to that identified in choroid plexus cells, where miR-204/211 keeps primed but quiescent neuronal stem cells to timely regulate cellular differentiation [45]. We believe that this process occurs through the direct control of Lmx1a on miR-204/211, since the ectopic overexpression of Lmx1a in non-neuronal cells such as HeLa is sufficient to drive miR-204/211 expression. However, other factors may also intervene in this process, since it has been recently shown that miR-204/211 expression can be regulated through epigenetic processes [63]. In addition, since, in the midbrain, Lmx1a/b drives the transition from the proliferative to the DA progenitor phase and the loss of Lmx1a/b activity results in the reduction of Th^+^ neurons during adulthood, we cannot rule out the possible involvement of Lmx1b in the process, although our data points to a clear role for Lmx1a.

Failure in miR-204/211 expression could result in an anticipated mDAn priming during development with a possible reduction of DA neurons in adult animals.

Indeed, several pieces of evidence point to the observation that, during development, neurogenic priming relies not only on the regulation of the translational repression complex elF4E1/4E-T that traps mRNA encoding for neurogenic transcription factors, but also on miRNA-mediated mechanisms [38,64].

Interestingly, most of the miR-204/211 targets we identified are directly linked to Nurr1. These include Gdnf [65], Elavl2 [66], Itpr1, Elavl4, Khdrbs1 [67], Rasgef1b [67], and Tcf12 [68]. Some of them are also involved in the differentiation of retinal progenitor cells and in the development of the eye (Nurr1, Elavl2, Plxna2, Rasgef1b, Wnt4, Sox11), tissues in which miR-204/211 shows a relevant role [42,44,66,69,70,71,72,73]

In a precise time window during embryonic development, miR-204/211 holds Nurr1 and its downstream targets repressed, preventing an early DA maturation. Later, during mDAn development, once the precursors are at the right differentiative stage, miR-204/211 expression becomes no longer necessary, Nurr1 expression is “released”, and the DA maturation can progress. In the case of miR-204/211 downregulation, early activation of Nurr1 would occur, resulting in premature DA differentiation and a consequent permanent reduction in the overall number of DA neurons. Our findings highlight the importance of fine-tuning miR-204/211 expression during development as an additional mechanism to control the timing of DA neuron differentiation.

We finally believe that our data are of relevance in the optic of human disorders. Indeed, the miR-204 expression has been reported as dysregulated in PD patients [39,40,41], and recently identified as a potential tumor suppressor in glioblastoma [74]. Thus, modulation of its expression could be relevant as a potential therapeutic approach.

## 4. Materials and Methods

### 4.1. Ethics Statement

Mice were bred in-house at the Institute of Genetics and Biophysics “Adriano Buzzati Traverso”, C.N.R., Naples, Italy. Selected animals were sacrificed in accordance with the recommendations of the national legislation as well as the European Commission (EU Directive 2010/63/EU for animal experiments). All the procedures involving mice were approved by the Ethic Scientific Committee for Animal Experiments (project id code: 491/2017-PR).

### 4.2. Tissues Collection

Mice brains were dissected under sterile conditions in PBS supplemented with glucose 33mM. Cortex and midbrain tissues at different ages were isolated under a stereomicroscope and processed for FACS sorting, RNA extraction, or primary cultures, as described below.

### 4.3. GFP^+^ Cell Sorting

Freshly dissected ventral midbrains from Th-GFP mouse embryos were enzymatically dissociated by incubation for 3 min at 37 °C in a trypsin solution (0.25% trypsin in 33 mM glucose/PBS; Sigma-Aldrich, Milan, Italy) containing 0.01% pancreatic DNAse (Sigma), and cell suspensions were sorted by BD FACSAria III into GFP-positive and -negative fractions. Cells were stored in RNAlater (Thermo Fisher Scientific, Milan, Italy) for further investigations.

Cell suspensions isolated from Pitx3-GFP were kindly provided by Prof. Marten P. Smidt.

Before FACS separation, ventral midbrains isolated from Pitx3-GFP and Lmx1a-GFP mice were dissociated using a papain dissociation system (Worthington, Milan, Italy) and MACS Neural Tissue Dissociation kit (Miltenyi Biotec, Bologna, Italy 130-092-628), respectively, as previously reported [75,76].

### 4.4. RNA Extraction and Real-Time qPCR

RNA was extracted from cells and tissues using the miRVana miRNA isolation kit (Ambion, Milan, Italy). The yield and the integrity of RNA were determined by spectrophotometric measurements, and agarose gel electrophoresis was used to determine the yield and quality of extracted RNA. One microgram of RNA was reverse-transcribed by 400 units of SuperScript III (Thermo Fisher Scientific, Milan, Italy) with 6 µM random hexamers (New England Biolabs) and RNAaseH (2U/µL; Thermo Fisher Scientific).

Real-time qPCR was performed on 2 µL of previously diluted cDNA (1:4) template, corresponding to 25 ng of original RNA amount, using the Power SYBR Green Master Mix (Thermo Fisher Scientific), in the presence of 0.5 µM of specific oligos. All analyzed genes shown in Table 1 were normalized vs. hypoxanthine phosphoribosyl transferase (Hprt). Analysis of relative expression of target genes was performed by the 2^−∆Ct^ or 2^−∆∆Ct^ methods.

### 4.5. TaqMan MicroRNA Assays

miRNA’s expression was quantified using TaqMan MicroRNA Assays (Thermo Fisher Scientific). Briefly, miR-204/211, miR-218, miR-9, and the reference snoRNA-202 were reverse-transcribed by the TaqMan MicroRNA Reverse Transcription Kit (Thermo Fisher Scientific) using specific primers for each transcript (TaqMan MicroRNA Assays). Then, qPCR was performed with TaqMan^®^ Universal Master Mix II, no UNG (Thermo Fisher Scientific) using the coupled primers and probe provided with the TaqMan MicroRNA Assays [Assay IDs: sno-202 #001232; miR-204/211 #000508; miR-9 #000583; miR-218 #000521].

### 4.6. Lentivirus Preparation and Viral Infection

cDNAs for mAscl1, mNurr1, and mLmx1a, as well as 275 and 340 base pairs encompassing pri-miRNA-204 and the pri-miRNA-218, respectively, were cloned into Tet-O-FUW or Tet-O-FUW-Ires-GFP lentiviral vectors under the control of the tetracycline operator. Lentiviruses were packaged in HEK293T cells as previously described [77]. Infections were performed in combination with rtTA transactivator viruses supplied with doxycycline (4 μg/mL; Clontech, Saint-Germain-en-Laye, France).

miR-204/211 and miR-218 were cloned into a PCR 2.1 TOPO TA cloning vector (Thermo Fisher Scientific) by using the following oligos flanked by the EcoRI recognition site: 204Fw: CCGGAGAATCAAGATGAGC; 204Rv: GTTATGGGCTCAATGATGG; 218Fw: GATCATACACAATCTGCGGGAAG; 218Rv: GGACATTTGTTATTCTCCCCTC.

### 4.7. Mesencephalic Primary Cultures (mE12.5-PCs)

Fresh single-cell suspensions obtained by the Trypsin-based enzymatic solution as described above were centrifuged 5′× 100× *g* and plated as in De Risi et al., 2021 [78] in a Neurobasal medium (NBM, Gibco, Milan, Italy), supplemented with B27 (Invitrogen, Milan, Italy), 2 mM L-glutamine (Gibco), penicillin, and streptomycin 10U + 10μg/mL (Pen/Strep, Sigma) with the addition of bFGF (20 ng/mL, Sigma), FGF8 (10 ng/mL, Sigma), and SHH (50 ng/mL; R&D systems, Abingdon-on-Thames, UK) at a density of 4 × 10^4^ cells/cm^2^ on poly-D-lysine (PDL) coated multiwells (15 μg/mL PDL for 1 h at 37 °C; Sigma).

After 3 days in culture, lentiviral vectors (LV) were added. At DIV6, the proliferative medium was replaced with the differentiating medium, NBM supplemented with B27, 2 mM L-glutamine, Pen/Strep, Ascorbic acid (200 μM, Sigma), 1 mM dibutyryl cyclic adenosine 3′, 5′-monophosphate (cAMP, Sigma), and 4 μg/mL of doxycycline to induce the expression of transgenes. Cell samples were collected at DIV12.

### 4.8. mes-c-myc-A1

The murine mesencephalic cells line mes-c-myc-A1 cells were obtained from the E11 midbrain and immortalized by the c-myc proto-oncogene. Cells were cultured in MEM/F12 (Gibco) supplemented with 10% FBS (Euroclone; Milan, Italy) and Pen/Strep at 1–2 × 10^4^ cells/cm^2^ density on PDL-coated plates. After 24 h, LV was added, and the expression of the inserted transcription factors was induced by adding doxycycline (4 µg/mL) to the medium. After 48 h following the infection, the medium was replaced by MEM/F12 supplemented with N2 (Invitrogen), 1 mM dibutyryl cyclic adenosine 3′, 5′-monophosphate (cAMP), SHH (50 ng/mL), FGF8 (10 ng/mL), and ascorbic acid (200 µM) and replaced every three days. Cells were harvested after 6 days.

### 4.9. Fluorescence-Activated Cell Sorting

GFP^+^ mes-c-myc-A1 cells were trypsinized, washed, sorted, or analyzed with BD FACSAria II (BD Biosciences, Milan, Italy). Cells were collected in RNAlater for RNA extraction.

### 4.10. epiSC Differentiation

epiSC were differentiated into mDAn as previously described [79]. Cells were plated in N2B27 medium (half NBM/B27 and half DMEMF12/N2) supplemented with bFGF and Activin (R&D) onto a 12-well plate coated with fibronectin (15 µg/mL in PBS, Millipore, Milan, Italy). After reaching semiconfluency, bFGF and Activin were withdrawn by the medium, and cells were infected. During the first two days of mDAn differentiation, the N2B27 medium was supplemented with 1 µM of the FGF/ERK inhibitor PD0325901 (Sigma) and then passaged onto PDL (15 µg/mL in PBS)/Laminin (20 µg/mL in PBS; Sigma) coated wells. SHH (200 ng/mL) and FGF8 (100 ng/mL) were added to the medium from day 5 (DIV5) to 9 (DIV9), while at DIV9 doxycycline (4 µg/mL) and ascorbic acid (200 µM) were added. Cells were cultured until day 14.

### 4.11. MEF and iDA Reprogramming

iDA neurons were generated from Mouse embryonic fibroblasts (MEFs) as previously described [36,78,80]. In brief, MEFs isolated from E14.5 mice embryos were infected with lentiviral particles coding for the TFs Ascl1, Nurr1, and Lmx1a in DMEM (Gibco) supplemented with 10% FBS, Pen/Strep, and doxycycline (4 µg/mL). After 48 h, the medium was replaced with DMEM/F12 supplemented with B27, Pen/Strep, and doxycycline until day 14.

### 4.12. Luciferase

For the putative miRNA-mRNA interaction, we used both TargetScan 7.2 (http://www.targetscan.org/vert_72/ accessed on 1 December 2021) and DIANA-microT-CDS (http://diana.imis.athena-innovation.gr/DianaTools/ accessed on 1 December 2021) [57,81]. Nurr1-3′UTR was cloned in the pMIR-Report Luciferase miRNA Expression Reporter Vector by using the following oligos Fw: CCAAGCACGTCAAAGAACT; Rv: ATCTCTAACTGTCGTACACC flanked by the SpeI and HindIII sites, respectively, for directional cloning. The miR-204/211 binding sequence was deleted by using the following oligos with the Quickchange site direct mutagenesis kit (Agilent, Rome, Italy): Fw: GTACATTGGAAAATCCTGACACACATAGTGTTTGTAACACCG; Rv: CGGTGTTACAAACACTATGTGTGTCAGGATTTTCCAATGTAC.

A luciferase assay was performed by using the Luciferase Reporter Assay System (Promega, Milan, Italy), following the manufacturer’s instructions. First, 400 ng of the pMIR-Report containing WT 3′UTR or mutated 3′UTR was co-transfected with 400 ng of the Tet-O-FUW-miR-204/211 and 400ng of the rtTA-expressing vector (400 ng) using Lipofectamine2000 (Invitrogen) in HeLa cells seeded the day before at a density of 40 × 10^3^ in a 48-well culture plate in DMEM, supplied with 10% FBS, Pen/Strep, 2 mM glutamine, 1 mM sodium pyruvate, and 100× non-essential amino acids. An empty pMIR-Report vector or Tet-O-FUW-miR-218 was used as further controls. Transfection efficiency was evaluated by the pRL-SV40 Renilla luciferase reporter vector (Promega). The firefly luciferase luminescent signal was normalized on the Renilla luciferase signal.

### 4.13. Putative miRNA-mRNA Interaction and Microarray Data Analysis

For the identification of miR-204/211 responsive elements within the 3′UTRs of differentially expressed genes in the Chabrat publication [56], we used DIANA-microT-CDS (http://diana.imis.athena-innovation.gr/DianaTools/ accessed on 1 December 2021) [57].

For the analysis of all miRNAs’ responsive elements enrichment in FP and NSC genes, we used ShinyGO v0.741 (http://bioinformatics.sdstate.edu/go/ accessed on 1 December 2021) with default parameters for miRNA.Target.TargetScan analysis [82].

Microarray data obtained for epiSC differentiated toward the DA phenotype and previously reported (GEO: GSE110270) [36] were analyzed to extract the fold-change values for Lmx1a, Nurr1, Th, miR-204/211, and selected genes specific for FP and NSC identity [45,58]. Venn diagrams were generated using InteractiVenn at http://www.interactivenn.net/ accessed on 1 December 2021 [83] while microarray values are represented by the ggplot2 library in R.

### 4.14. Statistical Analysis

The significance of differences for comparisons concerning the control was assessed by an unpaired t-test with Welch’s correction. A one-way ANOVA followed by Tukey post-hoc-test corrections was used for multiple comparisons, and a two-way ANOVA followed by Sidak post-hoc-test corrections was used for comparing DA with Ctrl samples among the differentiation time in vitro. The sampling dimension is reported for each graph. Analyses were performed using GraphPad Prism (GraphPad Software, San Diego, CA, USA).

## Figures and Tables

**Figure 1 ijms-23-06961-f001:**
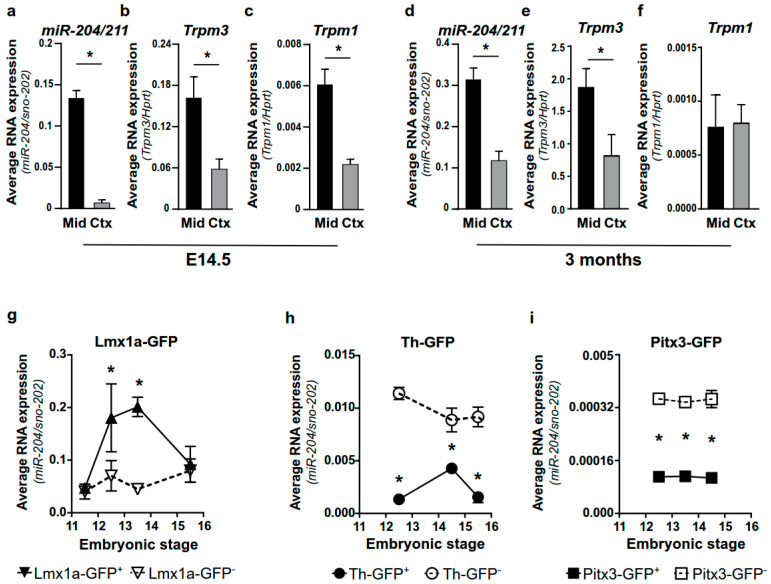
miR-204/211 is enriched in a DA precursor subpopulation during midbrain development. (**a**–**f**) TaqMan assay for miR-204/211 (**a**,**d**) and qPCR for Trpm3 (**b**,**e**) and Trpm1 (**c**,**f**) on microdissected midbrain (Mid) and cortex (Ctx) at E14.5 (**a**–**c**) or 3 months (**d**–**f**). miR-204/211 values are normalized to sno-202 expression, while mRNA levels are normalized on the reference mRNA hypoxanthine phosphoribosyl transferase (Hprt). Bars represent mean ± SD of 2^−ΔCt^ values from four animals. * *p* < 0.05 (unpaired t-test with Welch’s correction). (**g**–**i**) TaqMan assay for the expression of miR-204/211 in FACS-purified GFP^+^ and GFP^−^ cells from Lmx1a-GFP (**g**), Th-GFP (**h**), and Pitx3-GFP (**i**) reporter mice at different developmental stages. Data are normalized to the average of the reference sno-202 and represent the mean ± SD of 2^−ΔCt^ values from three independent experiments. * *p* < 0.05 of GFP^+^ with respect to GFP^−^ (two-way ANOVA followed by Sidak).

**Figure 2 ijms-23-06961-f002:**
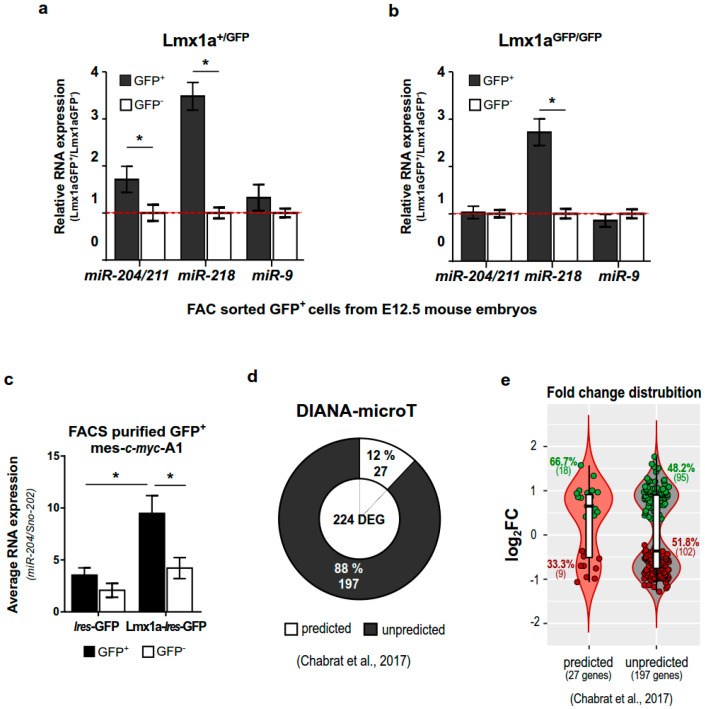
Lmx1a levels influence miR-204/211 expression and its repressive activity. (**a**,**b**) TaqMan assay for miR-204/211, miR-218, and miR-9 on FACS-purified GFP^+^ and GFP^−^ cells isolated from microdissected midbrains of Lmx1a^+/GFP^ and Lmx1a^GFP/GFP^ E12.5 embryos. miRNA values are normalized on the reference sno-202 and represent relative to control as mean ± SD of 2^−ΔΔCt^ values from three embryos. * *p* < 0.05 (unpaired t-test with Welch’s correction), (**c**) TaqMan assay for miR-204/211 on FACS sorted GFP^+^ and GFP^−^ mes-c-myc-A1 infected with LV-Lmx1a-Ires-GFP or LV-Ires-GFP. Values are normalized on the reference sno-202 and represent the mean ± SD of 2^−ΔCt^ values from three independent experiments. * *p* < 0.05 (one-way ANOVA + Tukey post hoc test). (**d**) Representation of the frequency of miR-204/211 target genes among the 224 differentially expressed genes (DEG) in mDAn knocked out for Lmx1a/b [56]. The percentage and number of genes are reported. DIANA microT-CDS was used to identify targeted genes (miTG score > 0.5). (**e**) Fold change distribution for predicted and unpredicted target genes of the miR-204/211 among the 224 DEG derived from Chabrat’s publication [56]. The percentage and number of genes are reported. Boxes in the violin plot represent the median, the 25th to 75th percentiles, and single values of log_2_ (Foldchange).

**Figure 3 ijms-23-06961-f003:**
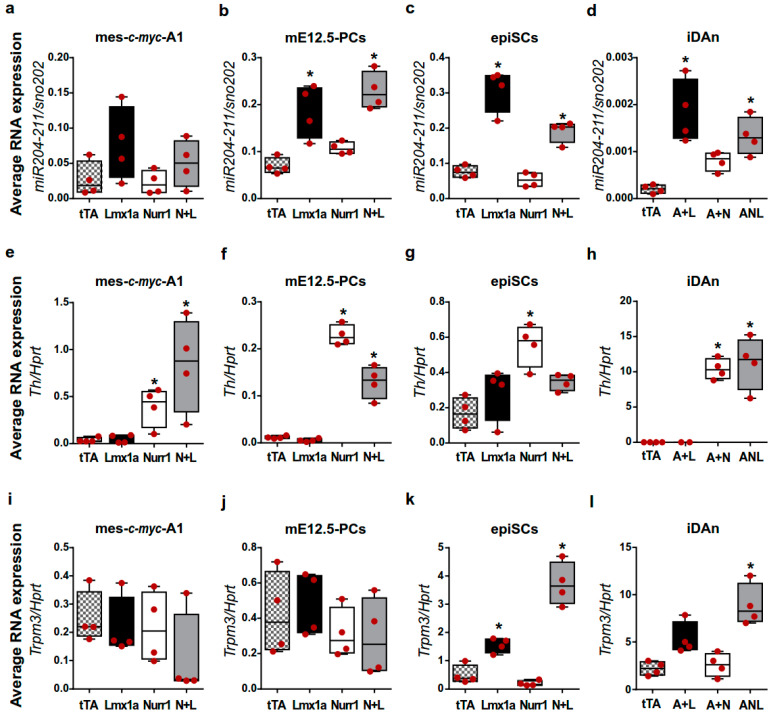
Lmx1a overexpression promotes miR-204/211 expression in vitro. (**a**–**l**) TaqMan assay and qPCR for miR-204/211 (**a**–**d**), Th (**e**–**h**), and Trpm3 (**i**–**l**) on differentiated mes-c-myc-A1 (**a**,**e**,**i**), mE12.5-PCs (**b**,**f**,**j**), epiSCs (**c**,**g**,**k**), induced dopaminergic neurons (iDAn; (**d**,**h**,**l**)) differentiated into mDAn. miR-204/211 values are normalized on the reference sno-202, while Th and Trpm3 levels are normalized on the reference mRNA Hprt. Data represent the median, the 25th to 75th percentiles, the minimum, and the maximum of 2^−ΔCt^ values from four independent biological replicates. L = Lmx1a, N Nurr1, A = Ascl1. * *p* < 0.05 with respect to controls infected with the only transactivator vector rtTA (one-way ANOVA + Tukey post hoc test).

**Figure 4 ijms-23-06961-f004:**
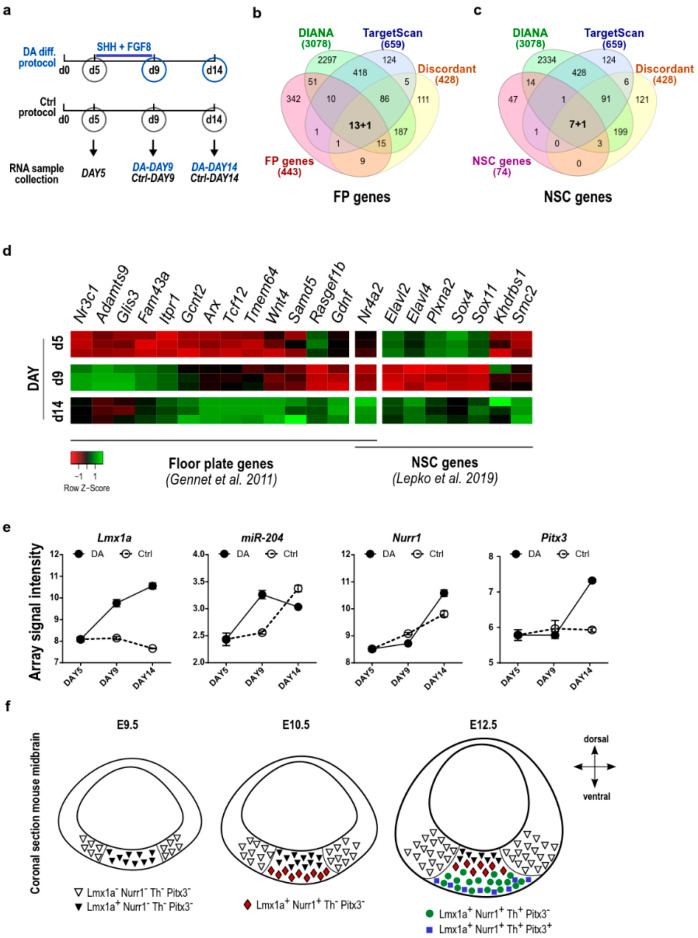
miR-204/211 regulates the expression of genes involved in mDAn differentiation. (**a**) Scheme of the DA differentiation protocol of epiSC. Samples from both DA and controls were collected in triplicate at day 5, day 9, and day 14. (**b**,**c**) Venn diagrams identifying miR-204/211 target genes (blue and green) among the floor plate signature (red) (**b**) and neural stem cells (NSC) signature (purple) (**c**) and showing discordant expression to that of miR-204/211 during in vitro DA-differentiation of epiSCs (orange). miR-204/211 target genes are identified by using both DIANA-microT-CDS and TargetScan. (**d**) Heatmap representing expression level during epiSCs DA differentiation [36] of miR-204/211 target genes selected among the floor plate (FP) and NSC signature genes [45,58]. Array values for a single sample are represented as colored boxes. Genes with reduced expression with respect to control are shown in red while genes with increased expression are shown in green. (**e**) Array data for Lmx1a, miR-204/211, Nurr1, and Pitx3 expression during DA differentiation of epiSCs. Data are represented as array signal intensity of DA-differentiated epiSC, and untreated cells cultured for the same time (see Section 4 for details). Bars represent the SD of triplicate values. (**f**) Schematic representation of Lmx1a, Nurr1, Th, and Pitx3 expression during mouse midbrain development.

**Figure 5 ijms-23-06961-f005:**
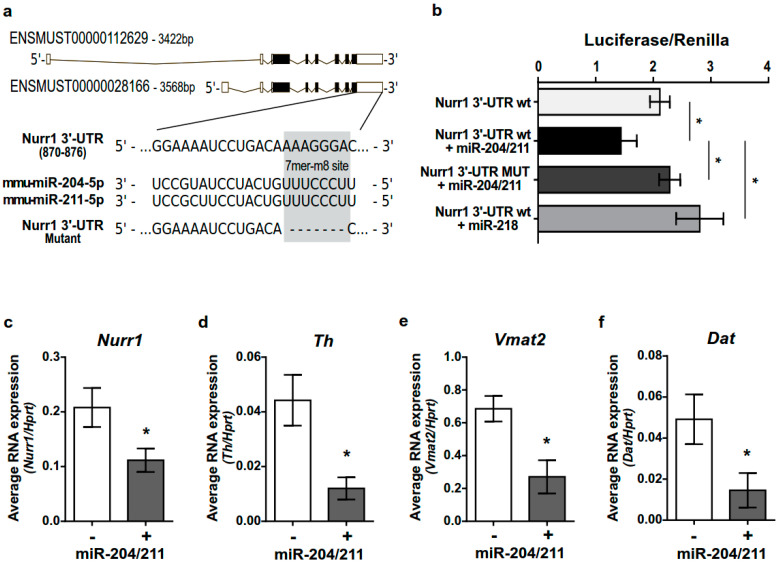
miR-204/211 targets Nurr1 3′UTR and its expression/activity in differentiating mDAn. (**a**) Schematic representation of miR-204/211 binding site on wild-type (wt) mouse Nurr1 3′UTR. The two existing Nurr1 transcript variants have the same 3′UTR and conserve an identical miR-204/211 binding site. The mutated sequence was also reported. (**b**) Luciferase assay on pMIR-Report construct containing the wt or MUT Nurr1 3′UTR downstream to the luciferase gene and transfected in HeLa cells, alone or in combination with miR-204/211 or miR-218. Data represent the mean ± SD of the ratio luciferase/renilla of three independent experiments. * *p* < 0.05 (one-way ANOVA + Tukey post-hoc). (**c**–**f**) qPCR for Nurr1, Th, Vmat2, and Dat on differentiated mes-c-myc-A1 overexpressing the miR-204/211. Data represent the mean ± SD of 2^−ΔCt^ values from three independent experiments. * *p* < 0.05 (unpaired t-test with Welch’s correction).

**Table 1 ijms-23-06961-t001:** Primer’s nucleotide sequence (5′-3′) used for qPCR.

Gene	Forward	Reverse
Dat	TCTGGGTATCGACAGTGCCA	GCAGCTGGAACTCATCGACAA
Hprt	TGGGAGGCCATCACATTGT	AATCCAGCAGGTCAGCAAAGA
Lmx1a	AACCAGCGAGCCAAGATGAA	TGGGTGTTCTGTTGGTCCTGT
Nurr1	CAACTACAGCACAGGCTACGA	GCATCTGAATGTCTTCTACTTAAT
Th	CCTTTGACCCAGACACAGCA	ATACGAGAGGCATAGTTCCTGAG
Trpm1	CTGCCTTGCTCAAAGGAACCAA	GAGGGGCCAGGCGGCCCAG
Trpm3	CATGCACTCCCACTTCATCC	TGGAACCCCTTGACCGATT
Vmat2	TTGCTCATCTGTGGCTGGG	TGGCGTTACCCCTCTCTTCAT

## Data Availability

Not applicable.

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
