# Peer review of "Lmx1a-Dependent Activation of miR-204/211 Controls the Timing of Nurr1-Mediated Dopaminergic Differentiation"

_ijms, 2022, doi:10.3390/ijms23136961_

Round 1

Reviewer 1 Report

The authors come up with ideas concerning an axis between the developmental expression of Lmx1a, the increase in microRNA (miR-) 204/211 and the regulation of orphan transcription factor Nurr1 which in turn controls the differentiation and maturation of midbrain dopaminergic neurons.

Using suitable reporter and knock-out systems authors show that Lmx1a is responsible for the upregulation of miR-204/211. By applying computational prediction tools authors found that miR-204/211 has a conserved binding site on the 3’UTR of Nurr1 transcripts. Using a luciferase reporter construct containing the miR-204/211 binding site authors show that miR-204/211 targets Nurr1 3’UTR and its expression/activity in differentiating mesencephalic dopaminergic neurons.

Authors detected a downregulation of Nurr1 by miR-204/211 affecting the expression of Nurr1 downstream target genes such as Th, Vmat2 and DAT. These data confirm the importance of fine-tuning miR-204 expression during development as a mechanism to control timing of DA neuron differentiation.

Major point: Although, authors cite the recent publication (39) by Wang, X.; Liu, L.; Zhang, L.; Guo, J.; Yu, L.; Li, T. Metab. Brain Dis. 2022, 37, 501–511, doi:10.1007/s11011-021-00866-6, findings are not reported properly in this manuscript: Wang et al have already described that NR4A1 (=Nurr1) acted as a target for miR 204-5p (= miR-204). Moreover, Wang et al  applied the starbase online tool and predicted that the NR4A1 3’UTR contained the potential 6-mer binding site of miR-204-5p (Wang et al 2022, Fig 5a) :

WT- NR4A1-3’UTR :         5’ gaGCG-GGGCUGGGAGGAAGGGAu 3’

miR-204-5p:                        3’  ucCGUAUCCUA…CUGUUUCCCUu 5’

Next, Wang et al 2022 verified a target relationship through a dual-luciferase reporter assay reporting that miR-204-5p prominently decreased the luciferase activity of WT- NR4A1 3’UTR group.

The 6-mer binding site in the 3’UTR of Nurr1 found by Wang et al 2022  may have an identical sequence portion as compared to the 7-mer claimed by the authors of this manuscript. Correct?  This point may be crucial for the manuscript and should be worked into the Introduction, the Results and in the Discussion sections. Specifically, the previous observation of the miR 204-5p mediated downregulation of Nurr1 protein expression as described by Wang et al should be mentioned as well.

Minor point: For the cited publication (38) Pereira, L.A.; Munita, R.; González, M.P.; Andrés, M.E. Long 3’UTR of Nurr1 MRNAs Is Targeted by MiRNAs in Mesence-658 phalic Dopamine Neurons. PLOS ONE 2017, 12, e0188177 authors may want to mention in the Introduction that transfection of rat mesencephalic neurons with mixed miR-93, miR-204 and miR-302d resulted in a significant reduction of Nurr1 protein levels.

Author Response

Response to Reviewer 1 Comments

Please find below, in red, detailed point-by-point responses to the reviewers’ comments.

The authors come up with ideas concerning an axis between the developmental expression of Lmx1a, the increase in microRNA (miR-) 204/211 and the regulation of orphan transcription factor Nurr1 which in turn controls the differentiation and maturation of midbrain dopaminergic neurons. 

Using suitable reporter and knock-out systems authors show that Lmx1a is responsible for the upregulation of miR-204/211. By applying computational prediction tools authors found that miR-204/211 has a conserved binding site on the 3’UTR of Nurr1 transcripts. Using a luciferase reporter construct containing the miR-204/211 binding site authors show that miR-204/211 targets Nurr1 3’UTR and its expression/activity in differentiating mesencephalic dopaminergic neurons. 

Authors detected a downregulation of Nurr1 by miR-204/211 affecting the expression of Nurr1 downstream target genes such as Th, Vmat2 and DAT. These data confirm the importance of fine-tuning miR-204 expression during development as a mechanism to control timing of DA neuron differentiation. 

Comments and Suggestions for Authors

Major Point 1: Although, authors cite the recent publication (39) by Wang, X.; Liu, L.; Zhang, L.; Guo, J.; Yu, L.; Li, T. Metab. Brain Dis. 2022, 37, 501–511, doi:10.1007/s11011-021-00866-6, findings are not reported properly in this manuscript: Wang et al have already described that NR4A1 (=Nurr1) acted as a target for miR 204-5p (= miR-204). Moreover, Wang et al  applied the starbase online tool and predicted that the NR4A1 3’UTR contained the potential 6-mer binding site of miR-204-5p (Wang et al 2022, Fig 5a) : 

WT- NR4A1-3’UTR :         5’ gaGCG-GGGCUGGGAGGAAGGGAu 3’ 

miR-204-5p:                        3’  ucCGUAUCCUA…CUGUUUCCCUu 5’ 

Next, Wang et al 2022 verified a target relationship through a dual-luciferase reporter assay reporting that miR-204-5p prominently decreased the luciferase activity of WT- NR4A1 3’UTR group.

The 6-mer binding site in the 3’UTR of Nurr1 found by Wang et al 2022  may have an identical sequence portion as compared to the 7-mer claimed by the authors of this manuscript. Correct?  This point may be crucial for the manuscript and should be worked into the Introduction, the Results and in the Discussion sections. Specifically, the previous observation of the miR 204-5p mediated downregulation of Nurr1 protein expression as described by Wang et al should be mentioned as well

Response 1: We thank the reviewer for this request that allows us to introduce better the orphan nuclear receptors and their involvement in DA neurons. Nur77 (NR4A1) and Nurr1 (NR4A2) are two different members of the orphan nuclear receptors family. They both have a role in DA neurons although with specific differences; while Nur77 seems mainly associated with dopaminergic transmission in mature neurons, Nurr1 is principally involved in the development of dopaminergic neurons.

The reviewer in his/her comment cites the miR-204/211 binding site on NR4A1 (Nur77) while our work is focused on the miR-204/211 binding site on NR4A2 (Nurr1). Thus, we believe there is no need to compare the 6-mer on Nur77 with the 7-mer binding site on Nurr1.

To this regard we can speculate that both Nur77 and Nurr1 host a binding site for miR-204/211 in their 3’UTR suggesting the importance of miR-204/211 regulation on DA neurons through either Nr4A1 or NR4A2.

Minor Point 2: For the cited publication (38) Pereira, L.A.; Munita, R.; González, M.P.; Andrés, M.E. Long 3’UTR of Nurr1 MRNAs Is Targeted by MiRNAs in Mesencephalic Dopamine Neurons. PLOS ONE 2017, 12, e0188177 authors may want to mention in the Introduction that transfection of rat mesencephalic neurons with mixed miR-93, miR-204 and miR-302d resulted in a significant reduction of Nurr1 protein levels.

Response 2: The paper from Pereira and collaborators has been clearly cited and described in the introduction (lines 109-111).

We would like to thank the reviewer for his/her constructive criticisms and insightful comments that helped us to improve our manuscript.

Reviewer 2 Report

Lmx1a dependent activation of miR-204/211 controls the timing 2 of Nurr1-mediated dopaminergic differentiation

Remarks to the Authors :

In manuscript from Pulcrano S et al., the authors, combining different techniques, identified the Lmx1a-miR-204/211-Nurr1 axis as a new pathway in the differentiation of midbrain dopaminergic neurons. This manuscript is well written and constructed and can be accepted for publication after the following comments being addressed.

Figure 1: Is not clear how Lmx1a regulates the levels of miRNA? Can the authors propose a mechanism?

Figure S1: the authors describe this mechanism also in HeLa cells. Is this new pathway not specific for the neurons? What are the effects on cancer cells?

Figure 4: going through the differentiation steps, the levels of Lmx1a increase while the one of miR-204/211 decrease (Figure 4e), is Lmx1b involved in any aspect of this mechanism? Is possible to quantify and predict the potency of the subpopulation here described? is this pathway deregulated in pathological conditions?

Author Response

Response to Reviewer 2 Comments

Please find below, in red, detailed point-by-point responses to the reviewers’ comments.

In manuscript from Pulcrano S et al., the authors, combining different techniques, identified the Lmx1a-miR-204/211-Nurr1 axis as a new pathway in the differentiation of midbrain dopaminergic neurons. This manuscript is well written and constructed and can be accepted for publication after the following comments being addressed.

Comments and Suggestions for Authors

Point 1: Figure 1: Is not clear how Lmx1a regulates the levels of miRNA? Can the authors propose a mechanism? 

Response 1: We thank the reviewer for his/her suggestion. We have expanded accordingly the discussion (lines 366-377) by addressing the possible regulatory mechanisms controlling miR-204/211 expression.

Point 2: Figure S1: the authors describe this mechanism also in HeLa cells. Is this new pathway not specific for the neurons? What are the effects on cancer cells?

Response 2: We thank the reviewer for his/her interesting questions. We used HeLa cells with the purpose to show that in a cellular context very different from that of a neuron the expression of Lmx1a is sufficient to upregulate miR-204/211. Interestingly, few reports address the role of miR-204/211 as a tumor suppressor able to interfere with cell cycle progression thus we speculate that the manipulation of miR-204/211 could result useful as a potential treatment for glioblastoma (lines 401-404). This is an emerging line of research that deserves future investigation.

Point 3: Figure 4: going through the differentiation steps, the levels of Lmx1a increase while the one of miR-204/211 decrease (Figure 4e), is Lmx1b involved in any aspect of this mechanism? Is possible to quantify and predict the potency of the subpopulation here described? is this pathway deregulated in pathological conditions?

Response 3: We thank the reviewer for his/her interesting comments/suggestions. Due to the high functional homology between Lmx1a and Lmx1b, we cannot rule out a possible regulatory role of Lmx1b on miR-204/211, although our data point to Lmx1a. This aspect is now properly discussed (lines 366-380).

In addition, to clarify the link between the deregulation of the Lmx1a/miR-204/211 pathway and human disorders we comment in the discussion on previous studies linking the variation of miR-204/211 expression to Parkinson’s disorders (lines 400-404).

We would like to thank the reviewer for his/her constructive criticisms and insightful comments that helped us to improve our manuscript.

Round 2

Reviewer 1 Report

Thank you for the clarifications concerning Nur77 and Nurr1.

For your information: Nurr1 and Nur77 may even be inversely regulated in cellular model systems, such as in human SH-SY5Y cells: DOI 10.1007/s12035-018-1311-6